# Patients’ and Relatives’ Preferences for a Palliative/Oncology Day Ward and Out-of-Hours Telemedicine—An Interpretive Description

**DOI:** 10.3390/healthcare9060758

**Published:** 2021-06-18

**Authors:** Eithne Hayes Bauer, Anders Nikolai Ørsted Schultz, Anette Brink, Lena Oechsle Jørgensen, Georg Bollig

**Affiliations:** 1Medical Research Unit, Institute for Regional Health Research, Hospital of Southern Jutland, University of Southern Denmark, 5000 Odense, Denmark; ANOS@rsyd.dk (A.N.Ø.S.); georg.bollig@rsyd.dk (G.B.); 2Department of Internal Medicine Sønderborg/Tønder, Hospital of Southern Jutland, 6200 Aabenraa, Denmark; Anette.Brink@rsyd.dk (A.B.); Lena.Oechsle@rsyd.dk (L.O.J.); 3Palliative Care Team, Department of Internal Medicine Sønderborg/Tønder, Hospital of Southern Denmark, 6200 Aabenraa, Denmark

**Keywords:** patient preferences, relative preferences, palliative care, oncology, day ward, telemedicine, interpretive description

## Abstract

Demographical challenges require adaptation and tailoring of services to suit palliative patients’ and relatives’ needs. Therefore, an interpretive descriptive study was performed to explore patients’ and relatives’ preferences for the establishment of a day ward and out-of-hours telemedicine. Semi-structured interviews were performed, and data were analysed using thematic analysis. Participants included patients (*n* = 12) and relatives (*n* = 5). Three themes emerged: (1) ‘Transport burden’ relates to transition from home-to-hospital-to-home and acknowledges the strain placed on patients and relatives. (2) ‘Role of relatives’ contemplates how the role of families in patient care influences patient preferences. (3) ‘Telemedicine—preferences and concerns’ covers preferences and concerns related to telemedicine in palliative care. The burden of transport and living alone play substantial roles in preferences for place of treatment. Relatives of palliative patients who avail of a day ward and telemedicine may experience an increase in the burden of care. Recognition of concerns pertinent to palliative patients and relatives is an important step in planning new services in palliative care. Concerns may be mitigated by rethinking referral guidelines, incorporating voluntary services, early integration of telemedicine into palliative care and examining patients and relatives’ expectations to care, but requires further research.

## 1. Introduction

Demographic changes are increasing the numbers of cancer and non-cancer patients requiring palliative care. Improved treatments, growing elderly populations and an increase in the number of multi-morbid patients are putting pressure on specialist palliative care services [1] which is also applicable to Denmark [2]. This trend is also reflected at the ward for palliative care at The University Hospital of Southern Jutland, where internal audits show that bed occupancy rate, referrals to the palliative team and staff overtime have been increasing steadily since 2017.

To address these changes, the managers of the medical department wish to increase service efficiency and maximise patient and caregiver satisfaction for patients admitted to the palliative/oncology ward. Previous studies support palliative patients’ interests in participating in telemedicine and display feasibility and an increased collaboration between both patients and families and specialists [3,4]. Therefore, inspired by a similar set-up in Scotland through the Highland Hospice [5], the plan is to establish a palliative/oncology day ward and out-of-hours telemedicine support for treatment and assessment of patients, where admission to the ward is not considered necessary, or is contrary to patients’ preferences. As a supplement to treatment on the day ward, patients will be offered out-of-hours telemedicine support. This service is not yet widespread in other palliative care settings in Denmark, and thus, knowledge in the area is limited. In order to apply a patient-centered approach to the establishment of the above-mentioned service and consider the preferences of palliative patients and their relatives, a study was commissioned.

Therefore, this study’s main objective is to shed light on patients’ and relatives’ preferences for the establishment of a day ward and out-of-hours telemedicine support for oncology and palliative patients. Apart from adding to the body of knowledge in the area, results will be incorporated into tailoring the service to suit the preferences of patients and relatives.

In the wake of the current Covid-19 pandemic, telemedicine is increasingly applied as a method of providing healthcare [6]. A paradigm shift has occurred, from an environment of care, where physical face-to-face contact was preferred, to an environment of care, where virtual contact without physical presence is acceptable for both patients and healthcare workers. Groundwork advancements have been accomplished in telemedicine that, prior to the pandemic, were decades underway [7]. Thus, telemedicine has proved its worth in many specialties. However, this study took place in autumn 2019 prior to the outbreak of COVID-19. Hence, the preferences expressed by patients and relatives are untainted by changes brought about by the imminent pandemic and, therefore, this study offers valuable insight into areas of concern to participants that are worth considering in post-pandemic hindsight.

The aim of the study is to explore patients’ and relatives’ preferences for the establishment of a day ward and out-of-hours telemedicine support for oncology and palliative patients.

## 2. Materials and Methods

In order to incorporate patients’ and relatives’ preferences into the service described above, a qualitative explorative study based on interpretive description was carried out among patients admitted to the ward and their relatives. Interpretive description was found appropriate for this purpose, as it supports the exploration and application of research in clinical settings [8]. Likewise, interpretive description recognises the merit of the researcher’s clinical background in appreciating clinical context and applicable results [9].

### 2.1. Setting

The Palliative/Oncology Ward at the Hospital of Southern Jutland, Denmark has 14 beds (2 oncology beds and 12 palliative beds) and provides oncological and palliative care and treatment for patients from four municipalities with 224,513 inhabitants [10]. Both cancer and non-cancer palliative patients are treated on the ward and two beds are reserved for oncology patients in curative treatment. A palliative/oncology day ward and out-of-hours telemedicine support was planned as part of a larger intervention using telemedicine to support patient’s and relative’s preferences for remaining in their own homes for as long as possible during their illness and to avoid unnecessary and repeated admissions [11]. The day ward will be physically located at the palliative/oncology ward with a capacity for four patients simultaneously. Nursing staff and a consultant from the ward will staff the day ward from 08.00–16.00. Previous experience with telemedicine for palliative patients in their homes has been found to contribute positively to the collaboration between both patients and relatives and palliative care specialists [3]. An application developed in Region Syddanmark and available to all inhabitants in the region called My Hospital (Mit Sygehus) will be employed to carry out telemedicine in the form of video consultations, which will be available during evenings, nights and weekends when the day ward is closed [12].

### 2.2. Data Collection

In total, 12 semi-structured interviews were performed with patients and relatives in autumn 2019 among patients admitted to the ward.12 individual interviews and 5 dyadic interviews—4 including 1 family member and 1 including 2 family members—were performed lasting between 25–57 min. The interview guide included open-ended questions pertaining to the aim and initial interviews tested the interview guide, which was found appropriate, as is recommended in interpretive description [9]. For further information, please see the interview guide included as Appendix A. Maximum variation was used to include as broad a range of patients as the study setting allowed; younger, elderly, living alone, living with relatives/spouses, male and female and distance from the hospital, permitting exploration of as many angles as possible, until data saturation was achieved [13]. Therefore, the only inclusion criteria was admission to the Palliative/Oncology Ward at the time of data collection. Exclusion criteria was documented cognitive impairment. Potential participants were recruited via a gatekeeper employed on the ward, who was neither in a managerial position nor a doctor involved in the treatment of recruited patients, thus, avoiding bias in relation to coercion into participation [14]. All interviews were transcribed verbatim.

## 3. Results

### 3.1. Participants

Participants (*n* = 17) comprised of 12 patients and 5 relatives; 8 men and 9 women in the age group 52 to 80 years of age with a median age of 73. Of the 12 patients included, 4 live alone and 8 live with a spouse or relative. Seven of the participating patients were declared terminally ill prior to inclusion. Despite maximum variation for gender, age, marital status and distance from the hospital, the majority of participants lived with relatives—see Table 1.

Overall preferences could be arranged into three groups; a preference for attending a day ward and out-of-hours telemedicine, a preference for admission to the ward and no interest in telemedicine and, finally, a preference for attending the day ward and no interest in telemedicine. Of patients living with spouses, 6 out of 8 expressed a positive attitude, 2 patients displayed preferences for a day ward without out-of-hours telemedicine support and all 4 patients living alone expressed a negative attitude towards a day ward and telemedicine.

### 3.2. Data Analysis

Data were analysed inductively using thematic analysis and interpretive description [15,16]. The first and second authors coded, categorized and themed all data until consensus was achieved. An excerpt of the analysis is presented in Table 2 below.

### 3.3. Themes

Three major themes and sub-themes were identified during the analysis. A short summary of each theme/subtheme will be presented. Relevant citations will be provided and themes will be described and interpreted. See Table 3 below for a brief overview of the themes.

#### 3.3.1. Transport Burden

When participants talked about transport, they referred not only to the physical act of transportation from a to b, but also to the time that it took to prepare and wait for transport, transport to the hospital, arrival and check in, waiting for an appointment, waiting for transport back home, transport home and settling in again at home. Almost all participants described transport as a burden. Certain situations could perpetuate the burden. Some preferred admission to the ward to avoid extra transportation, whereas others saw the day ward, and especially telemedicine, as a method of avoiding what they perceived as excessive transport in relation to the benefit involved.

“I would say that, if the treatment you have to receive, let’s say it takes 2 h or 3 h, well then, the day is more or less over.” (Participant 17, male relative, 63 years old).

Transport could be time-consuming for both patients and relatives, with some relatives having to take time off work to accompany patients to the hospital. If attending the day ward required frequent appointments, some patients living further away questioned the advantages of this for both patients and relatives, as it would increase the transport burden considerably and increase the amount of time spent away from home.

“I can’t exactly see how I could find the energy to go to a day ward. For example, I have no driving license, I have transport problems, maybe need to drive around for 3–4 h for a 2 h visit to the day ward. So, there’s something here that’s logistically wrong.” (Participant 14, male patient, 52 years old).

The alternative to relatives accompanying patients to the hospital could involve communal transport, but this is not without problems either. Often patients are required to travel together, which can increase the length of the journey. Furthermore, travelling together with strangers places an extra strain on the physical and psychological reserves of palliative patients, thus compounding the transport burden.

“It’s draining if you need to go out to the car and you’re not feeling well.” (Participant 9, male patient, 75 years old).

Patients described the physical exhaustion that they experience when forced to travel to and from hospital appointments. Hospital appointments could, therefore, contribute to a patient’s symptom burden due to transport.

“It takes pretty much a couple of hours to come back and forth. No, I dread it—that toing and froing!” (Participant 11, female patient, 78 years old).

One patient described how the thought alone of the length of time spent travelling back and forth to the hospital and the discomfort involved, was enough to fill her with anxiety.

“It’s crazy to drive to Copenhagen from here just to talk for 1 h and then home again. Many times you could take it on screen.” (Participant 6, male patient, 73 years old).

For some patients it made sense to avoid gruelling journeys to the hospital for appointments that could be carried out via telemedicine and they compared the scenario to situations from their working life, where long-distance meetings were carried out virtually. Previous experience with video conferencing helps to make sense of the situation and is seen as a method to reduce the transport burden.

“I don’t like waiting. I hate waiting. Yesterday they said that the ambulance would come within an hour—it came 2½ hours later. Super! That’s when Mrs. You-Know-Who starts getting a little hectic!” (Participant 8, female patient, 54 years old).

Longer transport times were described when travelling via communal transport. Prolonged waiting was particularly stressful, especially if patients were not informed of the same, or if waiting was prolonged unexpectedly. Because of their serious illness, patients felt ill-equipped to deal with these situations.

“It’s just as much the daily routines that get broken by having to get up early in the morning and leave and come home again in the evening, get ready for the evening and night at home and then again in the morning. I think that seems a little overwhelming for me. I prefer to be admitted to the ward instead of travelling to and fro, and to and fro.” (Participant 7, male patient, 76 years old).

Some patients mastered their illness by having daily routines. This predictability in their everyday lives helped them to cope with a serious illness. They described how the transport burden temporarily displaced their routines, which they found so difficult to cope with that they would prefer to be admitted to the ward, rather than be transported to the day ward for more than one day in a row. This was most prevalent in patients living alone.

“When you come home, there may be some things you just have to have under control. Maybe the bed needs to be fixed or something or other that you just don’t have the energy to do. Or there are things that need to be brought in so they’re close to you, so you don’t need to run all of the time.” (Participant 8, female patient, 54 years old).

Patients described settling in again after a trip to the hospital as the last part of the transport burden. Simple tasks which otherwise would not be taxing, were now an added burden, as they were mentally and physically drained on their return home. Again, patients living alone described this as a bigger burden thus, reiterating the importance of maintaining routines as a coping strategy for palliative patients, and how this can be affected by transport.

Palliative patients and relatives experience transport as a burden. Repeated or prolonged transport can be stressful for both parties. Patients possess various coping strategies for managing their illness, one of which is the importance of daily routines. Apart from the physical toil that transport puts on patients, it can also disturb their routines and, thereby, their coping strategies.

#### 3.3.2. The Role of Relatives

##### Importance of Relatives for Patient’s Preferences

Time is precious when living with a terminal illness. Most patients living with relatives—or even pets—were highly motivated to come home to their loved ones. They displayed a positive attitude towards the establishment of a day ward and out-of-hours telemedicine support. They believed that telemedicine could provide them with an opportunity to spend more time at home with loved ones. Patients living alone found the offer of a day ward and telemedicine less appealing than patients living with family did.

“I have a chihuahua at home. I love it more than anything and it can’t be in here in the hospital.” (Participant 8, female patient, 54 years old).

One patient poignantly described separation from her pet and how she was highly motivated to come home because of this separation.

“It’s not much fun lying here 20 of the day’s 24 h with nothing happening. We’ve been married for 52 years or something like that and we can’t forget her.” (Participant 15, male patient 74 years old).

“What if she only has to lie and wait here, then I would prefer to have her home.” (Participant 3, male relative 80 years old).

Patients living with relatives displayed a preference for the day ward with telemedicine. They described time spent during admission as time spent waiting, or time apart from loved ones and time that they would prefer to spend at home. Having someone to come home to is highly motivating and the day ward with telemedicine support represents a method of achieving this.

“I’m not able to myself—I don’t know how to work a screen” (Participant 9, male patient, 75 years old). “He doesn’t understand EDB (Electronic Data Processing—old-fashioned term for Information Technology (IT)), not really!” (Participant 10, female relative, 75 years old).

“No, no, not at all! But my wife definitely can, and the kids can.” (Participant 9, male patient, 75 years old).

Most relatives displayed a willingness to provide support in relation to using telemedicine. Thus, technically challenged patients living with relatives appeared less anxious about the use of telemedicine than did technically challenged patients living alone.

“Personally, I wouldn’t like it that much because I live alone. I wouldn’t like that with a screen. I’d feel lost… I think that, as a rule, I always prefer personal contact. I speak really well with the nurses who come home to me.” (Participant 1, female patient, 69 years old).

Patients living alone also displayed a wish to stay in their own homes but had a need for others to come and visit—either family members or primary healthcare workers. These visits could not be replaced by telemedicine. Fear of further social isolation and a need for physical closeness with other people outweighed a preference for the flexibility of a visit to the day ward and telemedicine support.

##### The Burden of Care—A Help or a Hindrance for Relatives

Several relatives displayed a willingness to assist patients with transport and telemedicine. Others pointed out that even if they could, they would not always do so, as they feel that the establishment of a day ward with telemedicine could increase the burden of care that already exists from supporting a seriously ill relative. Likewise, some patients also described concern over the added responsibility relatives could face from assisting with telemedicine, transportation and caring for them at home. Furthermore, they were worried on their relatives’ behalf about the lack of a possibility for respite from the burden of care that their admission to hospital would normally provide.

“I have travelled all over Europe and slept in many hotels and we’ve had really good beds and they served all sorts of peculiar things but sleeping at home in your own bed is always best...Your own smell and all that.” (Participant 8, female patient, 54 years old).

Despite the comfort of the palliative ward, patients prefer the surroundings of home for as long possible and home is something to strive for during their admission.

“It has taken its toll on our marriage—yeah, worn down because I’ve disturbed her all the time. As the doctor said, it wasn’t 100% necessary for me to come here. It’s just as much for us to come away from each other a little, so that we can come on the right course again.” (Participant 6, male patient, 73 years old).

For some patients, even if they wanted to come home, an admission to the ward may be beneficial for a relative or spouse at home, who needs a break from their caring role and this, in itself, may be reason enough for some patients not to avail of the day ward.

A potential dilemma exists for some patients and relatives; despite their wish to spend more time at home, relatives can end up having a heavier burden of care than before, which may result in the possibility of respite from the caregiver role being reduced.

#### 3.3.3. Telemedicine—Preferences and Concerns

##### Telemedicine—An Extra Comfort or an Added Worry

Several patients and relatives displayed a positive attitude towards the establishment of a day ward with telemedicine support. They described the importance of being at home and how telemedicine could play a role in achieving this goal. Familiarity, increased contact and reassurance in the ability to come into contact directly with specialists were appealing features. Patients also expressed concern about how telemedicine would be implemented.

“Sometimes you get pain some weird places. And so instead of going around and being afraid, you can get a hold of somebody straight away, I think that’s a good idea.” (Participant 4, female patient, 58 years old).

Telemedicine was described as having the potential to provide closer contact between specialists and patients, which is reassuring for patients especially in situations with unexpected reactions to treatment or increased pain.

“It’s good that it’s someone you know … I think that means a lot, that it’s someone you’re comfortable with.” (Participant 5, female patient, 68 years old).

Patients described that the idea of meeting a specialist with whom one is familiar, could increase feelings of confidence and comfort in relation to participating in telemedicine.

“That is, you feel a little more connected when you have a screen.” (Participant 10, female relative, 75 years old).

Others described that the ability to see and be seen by their specialist would be an advantage for them. They imagined that their contact with their specialist would improve, as a result of communicating via telemedicine thanks to the ability to see and observe the person you are talking to.

“I worked with revenue and yeah, I’m used to working with it. In the tax office, I worked with screens, so that’s no problem. I’d be able to have that contact.” (Participant 10, female relative, 75 years old).

Some relatives and patients displayed a preparedness to participate in telemedicine. Previous experience with computer technology gave them confidence in their own abilities and confidence in being able to assist their relatives, if the need should arise.

“You feel pretty quickly how good people are at it … if there are some eh, problems or signs of weakness, then I’m not sure that, but if I think that you’ve got a handle on it, then I have confidence in it.” (Participant 6, male patient, 73 years old).

In order to feel comfortable using telemedicine, patients described their expectations of a high level of professionalism amongst staff in how they master the technology involved. Likewise, lack of confidence in staff could have a negative impact on a patient’s preference for telemedicine and interaction with specialists. Ultimately, this could affect their preferences for treatment on a day ward with telemedicine support or admission to the ward.

Patients valued the potential of telemedicine; to help them spend more time at home, increase contact with specialists and improve familiarity and communication. Previous experience with technology from the workplace contributes to a positive attitude towards telemedicine amongst patients and relatives.

##### Screen Versus Telephone

Some patients described how they consider video consultations as superior to telephone consultations, whereas others prefer the simplicity of telephone communication.

“Of course I can say that I’m in a bit of pain, ah but it’s not that bad, but the moment you have a picture on, then you can see whether what I’m telling you is true. That’s not just something you can do through a telephone.” (Participant 8, female patient, 54 years old).

Patient assessment via telephone can be challenging and some patients describe downplaying symptoms. Patients described how they imagined that having a visual picture of a patient, as well as being able to hear them, could contribute to and improve the likes of pain assessment.

“We had that contact and I think that you feel that you’re a little bit closer together. That you have a little more contact than if it was just a telephone.” (Participant 10, female relative, 75 years old).

Again, being able to see one another during a conversation is described as creating more contact than a telephone can offer. The ability to evaluate non-verbal expressions and gauge non-verbal responses adds another dimension to encounters via video, which are not possible via telephone alone.

“Whether I can see them or not, it makes no difference.” (Participant 14, male patient, 52 years old).

On the other hand, some patients displayed no preference one way or the other—the most important thing was being able to get in contact with specialists when they needed them.

The type of contact that patients prefer is individual and it is important that they have the opportunity to choose. For some, telemedicine has the added benefit of being able to see whom you are talking to, whereas others find it makes no difference as long as they can get in touch when necessary.

##### Timing the Introduction of Telemedicine

While several patients had a positive attitude towards the establishment of a day ward with out-of-hours telemedicine support, some raised the subject of the timeliness of the introduction of telemedicine into a patient’s palliative trajectory.

“I think as a cancer patient you just have such a hard program with all sorts of treatments and that this would be just another disturbing element.” (Participant 14, male patient, 52 years old).

For some patients, dealing with their illness and all that it entails does not leave room for learning new communication skills, imagining that, rather than alleviate their situation, telemedicine would be an added burden.

“No. It wouldn’t work. Not for people who are so sick. I don’t think so.” (Participant 13, female relative, 56 years old).

“It should be the younger people who have it.” (Participant 12, male relative, 78 years old). 

“You need courage. For when you’re going to learn it.” (Participant 11, female patient, 78 years old).

Some questioned the appropriateness of introducing telemedicine in advanced palliative care, implying that it was too late in the course of their illness. They lacked both the necessary energy, and the interest in learning new skills. They suggested that due to the degree of their illness and their advanced age, telemedicine should be reserved for patients, who were younger, braver and not as ill.

“But yeah, I think that it’s a good idea that you try it out here on the ward because when you’re actually at home and you’re sick and it won’t do what you want it to do, then there’s a risk of, that there iPad will go flying through the air.” (Participant 8, female patient, 54 years old).

Others suggest that an introduction to telemedicine during hospital admission could be beneficial and make matters easier for when they returned home where technical problems may arise.

Overall, an earlier introduction to telemedicine, before patients reach the terminal phase of their illness, could result in more patients availing of a day ward and out-of-hours telemedicine later in their illness trajectory.

## 4. Discussion

Initial results revealed that the majority of the participants displayed a willingness to avail of the service. However, it became evident that the service was not perceived as one entity; some displayed an interest in a day ward but not in telemedicine, some displayed an interest in telemedicine but not in a day ward, some were interested in both and others again had no interest in either. This demonstrates the importance of tailoring services to suit the preferences of the individual and that one size does not fit all. Nonetheless, it is important to bear in mind that data were collected prior to the COVID-19 pandemic, which, as we now know, has had quite an impact on how we communicate and provide care within health sectors. On the other hand, the results of this study highlight important concerns among palliative patients and relatives about the establishment of new services—namely a day ward with out-of-hours telemedicine support. Several models and theories exist for examining the acceptance of telemedicine, one of which is the unified theory of acceptance and use of technology (UTAUT) [17,18]. UTAUT predicts acceptance based on performance expectancy, effort expectancy, social influence and facilitating conditions [17] and will be employed in the discussion to explain some of the results in relation to telemedicine.

The main findings identified in this study were divided into three themes. The first, referred to as ‘transport burden’, covers the physical and mental burden incurred by palliative patients as a result of transition between home and hospital. The second, entitled ‘role of relatives’, conveys the importance of family context for patient preferences from fuelling the desire to come home to concerns over additional encumbrances. The third, ‘telemedicine- preferences and concerns’, relates to patient and relative preferences for communication within the area of telemedicine.

### 4.1. Transport Burden

As we discovered in this study, transport is largely an unsettling event for palliative patients and their relatives. It covers a transition from one sector to another, which explains why transport is described as more than physical movement from a to b. Several phases were described in this extended transport or transition; preparation, transportation, waiting and settling. The combined effect of these phases could, therefore, be expressed in the manner of transport after-effects; physical and mental fatigue following a round trip to the hospital. According to Mezey et al., transitions at the end of life can be detrimental to patients and affect their quality of life [19]. Ingleton et al. also found that transport, and in particular ambulance transfer of seriously ill patients, was a core concern for patients and family carers and constituted a barrier in relation to facilitating choices for place of care and place of death [20]. This may help to explain why the majority of participants referred to transport as a burden and something to be avoided. WHO refers to transitions of care as; ‘the various points where a patient moves to, or returns from, a particular physical location or makes contact with a healthcare professional for the purpose of receiving healthcare [21]. Such a broad definition indicates how imprecise the concept of the transitions of care are. Nonetheless, WHO point out that transitions of care can be associated with adverse effects on patients, spanning from transient effects to increased mortality [21]. This makes it all the more important to recognise the effects that transport can have on palliative patients and their families and how establishing new services may lead to an increase in the transport burden on patients and families. A recent review of patient and provider satisfaction with telemedicine found that patient satisfaction was associated with less transport, reduced cost of transport and less time spent waiting [22], all of which were elements of the transport burden described in this study. Therefore, an awareness of the various phases of transport and what can be done to increase comfort, reduce waiting time and avoid unnecessary travel are important aspects that healthcare professionals and managers must take into account when planning treatment or appointments that require transport or the establishment of new services. This area warrants further investigation as it is poorly illustrated in the literature. This small explorative study has shown that transport is more than just physical movement from a to b for patients and, in fact, is of great concern to patients and relatives and warrants further investigation. Simple interventions such as avoiding appointments early in the morning to safeguard patients’ sleep, avoiding communal transport when possible, replacing physical appointments with virtual appointments, etc., can help to reduce the transport burden. One of the findings, maintaining daily routines as an important coping strategy for seriously ill patients living alone, is described in the literature as maintaining normalcy [23]. This strategy could be compromised by transport to and from a day hospital. Striking a balance between supporting autonomy and preserving dignity in this group of adults requires knowledge of factors affecting their care [23]. Advance Care Planning and telepalliative care may help to individualize care and maintain routines [24,25]. However, more research is necessary in order to examine how telemedicine can support older palliative care patients living alone.

The role of telemedicine in reducing or eliminating the transport burden is an important aspect of emergent palliative care and, in light of the current pandemic, has proven to be a useful asset to specialised palliative care [26]. Our study suggests that hospital appointments should be weighed against the transport burden that such a journey will incur and whether transport may further aggravate patients’ symptoms. Therefore, rethinking referral guidelines for palliative patients to strengthen the incorporation of telemedicine into standard specialised palliative care treatment is an important step, when creating patient-and family-centred approaches to address rising numbers of palliative care patients.

### 4.2. Role of Relatives

The Danish Health Board (Sundhedsstyrelsen (SST)) recommends that ‘palliative care is carried out individually and in cooperation with patients’ families’ [2]. Family involvement is considered one of the bedrocks of palliative care and families play a very important role in caring for palliative patients in the home. Previous studies have found that cohabitation is associated with dying at home and that family preferences play a role in facilitating home deaths [27,28]. Therefore, patient preferences will be influenced by relatives’ preferences and whether or not patients live alone or together with relatives will affect patient preferences. Knowledge of family context and background can help to provide valuable information about preferences for place of care and method of care helping to individualise services. SST also recommend flexibility in relation to place of treatment so that patients receive the best treatment to meet their needs [2]. This may involve treatment at home with telemedicine support, as preferred by several participants living with relatives, or an admission to the palliative care ward, as preferred by participants living alone. Palliative patients who lack a network are particularly vulnerable. O’Connor found that palliative patients living alone lose their social network and are afraid of becoming a burden, as their illness progresses [29]. Social networks are vital for assisting in home care and in order to maintain autonomy and remain at home, patients are willing to accept assistance [29]. Fear of losing their network is also evident among participants in this study and may explain why patients living alone displayed a preference for physical, rather than virtual, visits from palliative care specialists.

However, this finding is in conflict with the status for admissions to specialised hospital wards for palliative patients living alone. Adesrsen et al. found that the overall admission rate for patients living alone was lower than for patients living with spouses. The reason for this is unclear, but the hypothesis is that patients living with spouses have somebody at home who can advocate for their need for admission [30]. Hence, despite preferences for admission, patients living alone are less likely to be admitted to specialised palliative care facilities. Therefore, when deciding which patient groups will benefit most from a day ward appointment or an admission to the palliative care ward, it is important to consider whether they live alone, what social network they have, what their preferences are and how they are best supported.

Another important aspect to consider under the role of relatives is whether spouses will have an increased need for respite from the caregiver role or not, because of fewer admissions to the ward, qua attending the day ward with telemedicine support. An admission can be understood as serving a dual purpose—symptom relief for patients and respite for relatives. Therefore, screening relatives and recognising the need for respite will become more pertinent when establishing alternative services to admission. When considering family care in cancer settings, development of multimodal family interventions, including technology based interventions, are recommended [31]. The use of telemedicine to provide support to relatives and informal carers can play an important role. A screening tool such as The Carer Support Needs Assessment Tool Intervention (CSNAT-I) has been validated for use in the palliative care context [32,33]. It has recently been evaluated in a Danish context and has shown to have positive effects on caregiver distress, home care responsibility and caregiver’s experience of interaction with healthcare professionals [34]. Employing screening tools via telemedicine to examine families’ needs for support may help to avoid an increase in the burden of care that could arise as a result of the establishment of a day ward and telemedicine out-of-hours support. Furthermore, public palliative care educational interventions such as Last Aid and Online Last Aid can also play a role in empowering relatives and informal caregivers to teach them about basal palliative care, reduce feelings of helplessness and increase their understanding of the process of death and dying [35,36]. However, further research is necessary into carrying out both palliative support to carers and palliative care to patients via telemedicine.

In summary, it is important to bear in mind, that patients living with families and patients living alone have different preferences for care. Not all patients can, or will, avail of this service, as circumstances such as living alone can exacerbate feelings of social and physical isolation. While some will benefit from a trip to the day ward with out-of-hours telemedicine support, others will benefit more from an admission to the ward. A day ward and telemedicine can support patients’ living with relatives preferences for remaining in their own home. However, some relatives may experience an increase in the burden of care, which can be detected using screening tools. Telemedicine can be used to screen and provide support to family members.

### 4.3. Telemedicine—Preferences and Concerns

The rapid integration of telemedicine into palliative care during the Covid-19 pandemic provides further insight into the use of telemedicine in palliative care and anticipates the important role of telemedicine in post-pandemic palliative care [24,37,38]. Therefore, it is worthwhile considering palliative patients’ and relatives’ preferences and concerns in relation to the use of telemedicine in palliative care. Patients or relatives who possessed previous experience with video messaging or conferencing valued the potential of telemedicine as a method of spending more time at home, increasing contact with specialists, and improving familiarity and communication. If we consider these results in relation to UTAUT, a perceived acceptance of telemedicine is clearly present within this group, which can predict a behavioural intention to use [17]. Participants display performance expectancy through their expectations, in that telemedicine will help them to receive palliative care. Effort expectancy can be seen as they imagine ease of use through their familiarity with video contact. Social influence and facilitating conditions are present in the form of relatives/spouses support for the idea and they already possess the technology to participate in telemedicine [18]. All of the above lead to behavioural intention to use, which expresses this group’s acceptance towards the use of telemedicine. Viers et al. also describe how previous experience with video conferencing plays a role in preferences for participating in telemedicine describing previous experience as a positive indicator for participation in telemedicine [39]. This information can be translated into using algorithms to identify patient groups who are willing to participate in telemedicine or exposing groups that require further support and assistance [40]. This in turn may help to promote successful implementation of a day ward and out-of-hours telemedicine in palliative care.

As seen in this study, the type of contact that patients prefer is individual and it is important that they have the opportunity to choose. Some patients’ main concerns are not how to get in touch with specialists, but rather that they can get in touch when the need arises. Some patients in the study displayed a preference for face-to-face contact with specialists and thus performance expectancy—the idea that telemedicine would enhance contact or provide something useful for them—was not evident [18]. Patient autonomy in healthcare is based on informed consent and the principles of shared decision making, which can also be translated to telemedicine [41]. Promoting empowerment and supporting patients and relatives in choosing the type of contact that suits their preferences can ensure quality of care in telemedicine in palliative care, as well as standard palliative care. However, UTAUT does have its shortcomings when dealing with complex interventions in healthcare, as factors above and beyond the technology involved also play a role in telemedicine acceptance [18]. Willingness to participate in telemedicine in palliative care may also depend on how and when telemedicine is introduced into a patient’s illness trajectory. Some patients and relatives in the study perceived that introducing telemedicine in the terminal phase of illness could be a potential burden. Previous studies suggest that the early introduction of telemedicine in palliative care can lead to improved symptom management, comfort and patient and family satisfaction [42]. This iterates the importance of targeting telemedicine towards patients for whom it will not become an added burden, but at the same time, safeguarding patients’ autonomy and ensuring that, despite serious illness, they maintain the opportunity to choose a solution that is beneficial for them.

One method of introducing telemedicine at an earlier stage could be in the context of Advance Care Planning. Advance Care Planning (ACP) is a process of planning and documenting patients’ preferences for medical care together with patients, relatives and healthcare professionals so that it is ‘consistent with their values, goals, and preferences during serious and chronic illness’ [25]. Incorporating the option of receiving care via telemedicine into ACP conversations and documents may increase the level of commitment from healthcare workers—both in primary and tertiary settings—to respect and uphold patients’ preferences for telemedicine, telephone or face-to-face visits as their illness progresses. However, further research is necessary into the timeliness of the introduction of telemedicine into palliative care trajectories.

Another aspect to consider in relation to telemedicine is supporting patients who may wish to participate in telemedicine but do not feel equipped to do so. One method of ensuring extra support to patients and relatives in the implementation process is the use of volunteers. The Hospital of Southern Denmark under the Region of Southern Denmark has a strong collaboration with voluntary services which assist with various tasks from guiding patients and visitors with directions, to helping patients and relatives to download applications for communication with the hospital [43]. Combining healthcare provider and voluntary services can improve the quality of care, as is seen in areas such as hospice caregiver support and cancer centres [44]. Through combined efforts, it may be possible to reduce the transport burden, caregiver burden and implementation of telemedicine to provide patient-centred palliative care at a day ward with telemedicine support.

A willingness to participate in telemedicine depends on acceptance. Examining the level of acceptance of telemedicine in a group may provide useful information in relation to targeting initiatives to specific groups who require extra help and training. Actively incorporating the option of telemedicine into early palliative care in the context of ACP may increase successful implementation of telemedicine in palliative care.

## 5. Conclusions

This study aimed to explore patient’s and relatives’ preferences in relation to the establishment of a palliative/oncology day ward and out-of-hours telemedicine. Results indicate that while participants displayed a preference for attending a day ward and out-of-hours telemedicine, the burden of transport, relatives’ preferences and whether or not patients live alone play substantial roles in their preferences for place and method of treatment. Furthermore, relatives of palliative patients, who avail of a day ward and out-of-hours telemedicine, may experience an increase in the burden of care. Early integration of telemedicine into palliative care trajectories should be considered. Examining the level of acceptance of telemedicine in a group may provide useful information in relation to targeting initiatives to specific groups, who require extra help and training. Recognition of concerns pertinent to patients and their relatives and incorporating methods to mitigate these are important steps in planning new services in palliative care. Rethinking referral guidelines for palliative patients to day wards and telemedicine is necessary, along with further research into how the burden of transport and the coping strategies of adult palliative patients living alone affect patients’ and relatives preferences and expectations into place and method of treatment in palliative care.

## 6. Strengths and Limitations

A main strength of this study is the participation of terminally ill palliative patients and their relatives who contributed with data of immense forthrightness regarding their preferences for the establishment of a day ward and out-of-hours telemedicine.

Another strength is, that by carrying out an interpretive descriptive study the results are directly applicable to the clinical context of the study and can shape the establishment of a day ward and out-of-hours telemedicine.

A limitation of the study is that the setting comprises one centre only. Had the study been performed as a multi-centre, study results may have been different, as the culture of the setting can affect the results. Further research is necessary in order to determine further preferences of patients and relatives.

## Figures and Tables

**Table 1 healthcare-09-00758-t001:** Demographics of patients and relatives participating in a study on preferences in relation to establishing a palliative/oncology day ward with out-of-hours telemedicine support.

Participants *n* = 17	Gender F/M	Age	Patient Relative P/R	Living Alone or Living with Relatives	Distance in km from Hospital to Patient’s Home Towns
1	F	69	P	Alone	70
2	F	79	P	With spouse	2
3	M	80	R	With spouse	2
4	F	58	P	? ^1^	65
5	F	68	P	Alone	40
6	M	73	P	Alone	65
7	M	76	P	Alone	15
8	F	54	P	With spouse	45
9	M	75	P	With spouse	70
10	F	75	R	With spouse	70
11	F	78	P	With spouse	70
12	M	78	R	With spouse	70
13	F	56	R	With relatives	70
14	M	52	P	With spouse	40
15	M	74	P	With spouse	50
16	F	63	P	With spouse	70
17	M	63	R	With spouse	70

^1^ missing data.

**Table 2 healthcare-09-00758-t002:** An excerpt of data analysis of patient and relative preferences in relation to establishing a palliative/oncology day ward and out-of-hours telemedicine support.

Main Theme	Category	Subcategory	Code	Citations
Transport burden	Frequent trips increase transport burden	A burden for patients and families. Transport time and waiting time are time consuming. Time disappears and other jobs remain undone. Dilemma—relatives want to accompany patients but feel a lot of time spent waiting. Frequent trips to day ward disadvantage.	Time consuming, waiting for treatment, time wasted, long journey,	“I would say that, if the treatment you have to receive, let’s say it takes 2 h or 3 h, well then, the day is more or less over.” (Participant 17, male relative, 63 years old)
	Communal transport adds to transport burden	Alternative to relatives transporting patients. Communal transport takes longer. Requires extra energy from patients. Travelling with strangers extra strain on the physical and psychological reserves. Increases transport burden. Long transport time for short visit.	Transport issue, lack of resources, communal transport tougher for patients, energy draining, strangers/other patients	“I can’t exactly see how I could find the energy to go to a day ward. For e.g., I have no driving license, I have transport problems, maybe need to drive around for 3–4 h for a 2 h visit to the day ward. So, there’s something here that’s logistically wrong.” (Participant 14, male patient, 52 years old)
	Reducing transport burden	Telemedicine can replace some appointments. Makes sense Comparison to previous work practice. Experience with video conferencing. Alternative to physical appointments. Avoid journeys.	Making sense, Previous experience, Telemedicine, Method to reduce transport burden	“It’s crazy to drive to Copenhagen from here just to talk for 1 h and then home again. Many times you could take it on screen.” (Participant 6, male patient, 73 years old)

**Table 3 healthcare-09-00758-t003:** Themes, sub-themes and main points in a study on preferences in relation to establishing a palliative/oncology day ward with out-of-hours telemedicine support.

Themes	Sub-Themes	Main Points to Be Discussed
Transport burden	Transition from home-to-hospital-to-home	Transport places a strain on patients and families involving more than the physical act of transportation from a-to-b.
Role of relatives	Importance of relatives for patient preferences	Whether or not patients live alone affects patient’s preferences for care. Relatives’ preferences can affect patients’ preferences for care.
	Burden of care—a help or a hindrance	A day ward appointment and/or telemedicine instead of admission may lead to lack of respite for relatives. Relatives’ play a role in providing technical support.
Telemedicine—preferences and concerns	Telemedicine—an extra comfort or an added worry	Increases the opportunity for patients to remain longer in their own home. Previous experience with video communication contributes to a positive attitude to telemedicine. Implementation is a cause for concern. Staff proficiency in handling telemedicine may affect patient preferences.
	Screen versus telephone	The visual aspect of telemedicine is preferable in comparison with talking on a phone. The method of communication is less important than the ability to gain access to specialists when necessary. Some patients prefer the simplicity of telephone communication.
	Timing the introduction of telemedicine	An earlier introduction to telemedicine in the illness trajectory would be preferable.

## Data Availability

Permission to store data was received from The Region of Southern Denmark—case number 19/41687, 27 August 2019. All data were anonymised following transcription and stored securely.

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
