# Peer review of "Patients’ and Relatives’ Preferences for a Palliative/Oncology Day Ward and Out-of-Hours Telemedicine—An Interpretive Description"

_healthcare, 2021, doi:10.3390/healthcare9060758_

Round 1

Reviewer 1 Report

The aim of the paper is to examine patients’ and relatives’ views on the establishment of a palliative/oncology day ward and out-of-hours telemedicine. Its main contribution is that the information was gathered directly from those experiencing the phenomenon under investigation, emphasizing on the literal description and further interpretation of the findings.

The authors clearly stated the aim of the research and its relevance. Nevertheless, the objective could be revised regarding to the word “explain” and considering the word “explore” instead; this would be congruent with Line 72: “Interpretive description was found appropriate for this purpose, as it supports the exploration”, and with Line 94 “…permitting exploration of as many angles.” On the other hand, and citing Bradshaw and cols. (2017), …the goal of qualitative description research is not “discovery” as is the case in grounded theory, not to “explain” or “seeking to understand” as with ethnography, not to “explore a process” as is a case study or “describe the experiences” as is expected in phenomenology”, the aim may be to “describe” the worldviews of the people involved.

This reflection conducts to comment about the research question, which is not clearly formulated, and regarding the patients’ and relatives’ “views”. The authors used different terms all over the manuscript that confuse the reader, are they describing the views, the preferences, the concerns, the wishes, the need, the perspectives, or the expectations? If each term means the same or different concepts it must be stated. A suggestion to consider preferences as it is the most used term (preferences (38); concerns (20); Views (13); needs (12); acceptance (12); attitude (7); wishes (3); expectations (3)). But that would depend on what or how did you build the questions for the interview. The manuscript must show congruency.

The introduction section of the paper would be benefit if the authors add some evidence as an example of the increasing requirement of palliative care at the university hospital where the study was held. It is also suggested to include the definition or description of a “palliative/oncology day ward” and “out-of-hours telemedicine support”, to be clearer for global readers.

On the Methods section it is clear how the participants were selected, and included a range of experiences (age, gender, marital status, severity of condition). The convenience sample allowed the researchers to select available participants. Data were collected by semi-structured interview, but to ensure the possibility to repeat the study, authors must add information like interview guides, duration of the interview, number of interviews per participant, if the methods were modified and why (transferability and dependability). Authors did not mention if they had appropriate referral system given the possibility to evoke emotions on participants. There is no information of the establishment of rapport, trusting relationship and empathy during the interview (credibility).

Data analysis is well described and rigorous, the analysis process is demonstrated on table. It is suggested to add another table to show the themes and subthemes. Findings represent the data gathered, evidenced by the inclusion of direct quotations of participants (confirmability), but there is no information about triangulation as example. In the discussion the authors explain how the findings can be transferred or compared to other studies and populations as well as its contribution to the existing knowledge. Conclusions must show the answer to the research question. Remember that some readers are just going to read the abstract and the conclusions. Authors must be clear about the conclusion leaded by the participants and the descriptions they made about their (views or preferences). New areas of research are identified, and recommendations offered.

Typos

line 41 “the plan is to establish a palliative…” or “they plan to establish a palliative…”

Bradshaw, C., Atkinson, S., & Doody, O. (2017). Employing a qualitative description approach in health care research. Global qualitative nursing research4, 2333393617742282.

https://journals.sagepub.com/doi/full/10.1177/2333393617742282#bibr15-2333393617742282

Author Response

Dear Reviewer 1,

Firstly, I would like to thank you very much for your review and for your valuable comments, which contribute to the improvement of this manuscript.

Attached is a document containing a table with your comments, my response and subsequent amendments. Changes to the manuscript are highlighted.

All authors have read and approved the included amendments.

Yours sincerely,

Corresponding author.

Reviewer 2 Report

Thank you for the opportunity to review Manuscript ID healthcare-1208399, entitled "Patients’ and Relatives’ Views on a Palliative/Oncology Day Ward and Out-of-Hours Telemedicine – An interpretive description.”

This article provides insight into the facets of how patients and relatives view the establishment of a palliative/oncology day ward and after hours support via telemedicine. The study was conducted well before the COVID-19 pandemic which shaped telemedicine in unforeseen ways. The authors discover the three main themes “transport burden”, “role of relatives” and “telemedicine – preferences and concerns” of which the theme “transport burden” seems to play the most prominent role (together with the positive effect of telemedicine here).

Relating their qualitative findings to the demographic data at hand, the authors open new perspectives on factors of acceptance or refusal of day ward or telemedicine support amongst patients and relatives.

Over all I think this article is suitable for publication, it is well written and presents new findings which are – in light of the new influences of the COVID-19 pandemic on telemedicine – even more interesting.

Therefore I recommend publication.

I have very few and very minor remarks:

Line 115: is the average a mean or a median? Median makes more sense in my opinion.

Line 121: I think it should read “...attending the day ward”

Line 158: I suggest caution with the term of “unnecessary” transport because it is not entirely clear for me who shaped it. The Patient? Relative? Physician? I think the authors report the patients’ expression of “unnecessary” transport meaning something like “too little benefit for the burden”. This may be expressed with a little more clarity.

Line 217-234: this is an excellent point in my opinion - it may be covered a bit more in the discussion.

Line 656: typo, frequently seen: it should read “Advance Care Planning”

Author Response

Dear Reviewer 2,

Firstly, I would like to thank you very much for your review and for your valuable comments, which contribute to the improvement of this manuscript.

Attached is a document that contains a table with your comments, my response and subsequent amendments. Changes to the manuscript are highlighted.

All authors have read and approved the included amendments.

Yours sincerely,

Corresponding author.

Reviewer 3 Report

The authors describe a trend in healthcare which was slow at the time of their study and came in to a momentum with the COVID pandemic. The authors integrated this well.

Although there is (too) much text some things remain unclear:

  • Who are these patients? You mention oncological and palliative treatment: palliative antitumor treatment or also symptom control? I think to understand the former, but in the conclusion you speak of terminally ill patients. If you mean terminally ill patients: why have them on the ward anyway?
  • The initiative was from the managers: sounds money-driven: please explain: what was the real intention: patient and caregiver satisfaction or efficiency (nothing wrong with both but the intention is important)
  • Maybe it would help if we could see the questions asked

The paper has to be restructured and shortened. For example:

  • In data analysis: “All authors took part in study design and discussion of results.” doesn’t belong here
  • “Participants are presented below and participant demographics are outlined in Table 1. Results are presented in themes and sub-themes. Findings are presented with supporting citations followed by interpretive descriptions. Table 2 presents an example and excerpt of the method of analysis employed.” Is double with what comes next and confusing
  • “Despite maximum variation for ………the majority of participants were women..” 9 out of 17
  • “Overall preferences could be arranged into three groups….” Please add day for ward in the last option.
  • Table 2 is an example of the excerpt you made? I presume you made them after identifying the three themes?
  • Line 236: just give the conclusion of part one, do not interpret here: belongs in the discussion. And to conclude here that it could influence their coping is much overdone.
  • Line 320: again: time consuming: we saw that already
  • Line 342: interpretation: belongs in the discussion
  • Line 355: “Previous experience of video conferences….”we saw that already: double
  • Line 360 – 368: same as 313 and further
  • Line 395..: double
  • Line 415: phone versus screen: we saw that already in line 383
  • Discussion: be careful to mingle transition and transport: for me it is unclear whether you mean another place or another phase with transition.
  • In conclusion: “Algorithms to screen patients in relation to 684 preferences and use of shared decision-making may help to individualize place of 685 treatment.”??? Let’s just ask them! You showed there are many different preferences

Overall: interesting but way to extensive: please describe the essentials

Author Response

Dear Reviewer 3,

Firstly, I would like to thank you very much for your review and for your valuable comments, which contribute to the improvement of this manuscript.

Attached is a document which contains a table with your comments, my response and subsequent amendments. Changes to the manuscript are highlighted.

All authors have read and approved the included amendments.

Yours sincerely,

Corresponding author.

Round 2

Reviewer 3 Report

No further comments